# Computational Tools to Rationalize and Predict the Self-Assembly Behavior of Supramolecular Gels

**DOI:** 10.3390/gels7030087

**Published:** 2021-07-09

**Authors:** Ruben Van Lommel, Wim M. De Borggraeve, Frank De Proft, Mercedes Alonso

**Affiliations:** 1Molecular Design and Synthesis, Department of Chemistry, KU Leuven, Celestijnenlaan 200F Leuven Chem & Tech, P.O. Box 2404, 3001 Leuven, Belgium; wim.deborggraeve@kuleuven.be; 2Eenheid Algemene Chemie (ALGC), Department of Chemistry, Vrije Universiteit Brussel (VUB), Pleinlaan 2, 1050 Brussels, Belgium; fdeprof@vub.be

**Keywords:** supramolecular gels, LMWG, computational chemistry, molecular dynamics, modeling, self-assembly

## Abstract

Supramolecular gels form a class of soft materials that has been heavily explored by the chemical community in the past 20 years. While a multitude of experimental techniques has demonstrated its usefulness when characterizing these materials, the potential value of computational techniques has received much less attention. This review aims to provide a complete overview of studies that employ computational tools to obtain a better fundamental understanding of the self-assembly behavior of supramolecular gels or to accelerate their development by means of prediction. As such, we hope to stimulate researchers to consider using computational tools when investigating these intriguing materials. In the concluding remarks, we address future challenges faced by the field and formulate our vision on how computational methods could help overcoming them.

## 1. Introduction

Supramolecular gels, often referred to as molecular gels or low molecular weight gels (LMWGs), are a type of soft material that mostly consists of two constituents: a solvent, which accounts for up to 99% of the material, and a small molecule termed a gelator. Although only present in small amounts, the gelator provides the material with the typical viscoelastic gel properties, by forming a self-assembled network that spans and immobilizes the solvent [1,2,3,4]. While there is a large chemical diversity among reported gelators, most of them rely on the ability to form intermolecular noncovalent interactions in an anisotropic fashion. Hence the term supramolecular gels. In the past decade, the field of supramolecular gels has reached its adolescent years, providing researchers with ample well-established experimental techniques to characterize and investigate these materials, amongst others rheological measurements, various microscopy imaging techniques, NMR spectroscopy, UV-VIS spectroscopy, etc. [5,6,7,8]. Evidently, these experimental techniques have played a critical part in establishing our current understanding and development of supramolecular gels and will remain essential in the forthcoming years.

However, the macroscopic properties of these soft materials are governed by weak noncovalent interactions. Gaining direct insights into these interactions via wet-lab experiments is challenging. Fortunately, in the past, various computational methods and tools have proven their usefulness when investigating noncovalent interactions in supramolecular materials [9,10,11]. In this review, we set out to create awareness on the use of computational chemistry within the field of supramolecular gels. The first part discusses how computational methods can help to improve our understanding of supramolecular gels, often complementing the observations from experimental techniques. Next, several recent computational studies aiming to accelerate the development of supramolecular gels by a priori predictions are highlighted. In the concluding section, we look forward towards the opportunities and challenges that remain in the field of supramolecular gels where computational methods can be of considerable value.

## 2. Rationalizing Supramolecular Gelation

Gaining a deeper understanding of supramolecular gels is not only useful for scientific purposes, but it can also aid in their development and increase their applicability. To this end, several computational techniques have been applied to acquire knowledge about the conformational preference of gelators in a solvent environment, possible stacking modes of the nano-architectures, the noncovalent interactions in and between gelator molecules and additives, etc. Having said that, supramolecular gelation is a multiscale problem, with interactions occurring at the atom scale governing macroscale material properties [2,12]. As a result, different computational techniques with different levels of accuracy are required to gain information at different scales (Figure 1). In this section, an overview of computational methods employed to rationalize the gelation behavior is provided. While this section is organized based on the scale and level of accuracy of the computational method, we do mention that in order to obtain the most complete picture, a combination of these techniques integrated in a multiscale approach is desirable [13,14,15].

### 2.1. Static Quantum Mechanical Calculations

Computations performed with techniques that rely on quantum mechanics, generally yield highly accurate properties. Even within quantum mechanical methods, subdivisions of techniques can be made based on the manner in which electron correlation effects are taken into account [16]. This means, however, that they also require a significant amount of computational power. Hence, quantum mechanical calculations in the field of supramolecular gels are currently limited to static or short dynamic simulations of single gelator molecules or systems of up to approximately 500 heavy atoms. Additionally, solvent effects (if any) are included by implicit continuum models or explicitly by molecular mechanics methods (QM/MM). Among the many quantum mechanical theories available, density functional theory (DFT) has established itself as the primary quantum mechanical workhorse to investigate supramolecular systems [17,18,19]. The popularity of DFT can be attributed to its favorable balance between accuracy and computational cost, its flexibility (i.e., a myriad of exchange correlation functionals are developed to address different problems) and its ease of use (i.e., several user-friendly software packages exist, to set up a DFT calculations in a matter of seconds). Nevertheless, it is well known that conventional DFT functionals fail in describing long-range dispersion interactions, which are crucial to accurately model the noncovalent interactions in a supramolecular gel [20,21,22]. Luckily, numerous methods to improve the description of long-range dispersion interactions have been developed, including long-range corrected functionals [23,24], atom-pairwise additive schemes [25], or non-local dispersion corrections [26,27]. It goes without saying that, when studying supramolecular gels using DFT, attention must be paid to accurately describe long-range dispersion effects in order to obtain meaningful results.

In 2008, Urbanová and co-workers were one of the first to obtain structural information of a supramolecular gel from DFT calculations [28]. In their work, the structure of a guanine derived supramolecular hydrogel was investigated by systematically computing stable conformations of a simplified model of the gelator, ranging from dimers to tetramers. By comparing the DFT calculated VCD and IR spectra with the corresponding experimental spectra of the hydrogel, a plausible supramolecular stacking model was put forward. The proposed stacking model relied on Hoogsteen base pairing hydrogen bonds between the gelators [29]. Importantly, only a good agreement with the experimental spectra could be obtained by adding a sodium ion to the structure. The researchers rationalized that the presence of the sodium ion could stem from the pH-triggered gelation procedure (Figure 2). Following this study, other groups relied on DFT calculations to suggest a reasonable supramolecular stacking of gelators, from which both structural and electronic information could be retrieved [30,31,32,33,34]. However, the computational requirements of the quantum mechanical DFT calculations limit the size of the system that can be tackled. Additionally, when the complexity of the molecular system increases, conformational space increases as well. For this reason, it is recommended that such computational studies consider only a simplified model of the gelator and are supported by experimental data. Apart from FT-IR and VCD spectra, the experimental UV-VIS spectra can be compared with the absorption spectra obtained from time-dependent DFT calculations (TD-DFT), strengthening the validity of the proposed stacking model [35,36]. In a more recent example, Zwijnenburg and co-workers relied on TD-DFT calculations to gain insight into the self-assembly and gelation of an alanine functionalized perylene bisimide gelator [37]. First, stable aggregates with different sizes (up to trimers) and protonation states were optimized at the DFT level. Subsequently, they obtained the respective TD-DFT computed UV-VIS spectra of the different structures. With these spectra, they were able to assign the changes in the experimental UV-VIS spectra during the glucono-δ-lactone pH-triggered gelation to a particular stacking of gelator molecules.

The accuracy that can be attained with DFT calculations goes hand in hand with significant computational requirements and size limitations. However, it has been shown that even calculations on a single gelator molecule, or a simplified model thereof, could provide valuable insights into the self-assembly behavior [13,38,39,40]. In 2013, Xie and co-workers were able to rationalize the gelation performance of a set of small pyridyl urea-based gelators through DFT calculations at the single gelator level [41]. Owing to the molecular simplicity of the gelator, they could perform a full conformational analysis on these structures. Surprisingly, when the pyridyl ring of the gelator was substituted in an *ortho*-fashion, conformations with an intramolecular hydrogen bond became accessible. These intramolecular interactions compete with the otherwise intermolecular urea hydrogen bonding, which is crucial for the self-assembly process. They rationalized that these observations could explain the drastic decrease in gelation performance when changing the substitution pattern of the pyridyl ring from *para* to *ortho* (Scheme 1). In addition, Wezenberg et al. showed the importance of intramolecular interactions on supramolecular gelation [42]. In their work, a photoswitchable urea-based gelator was developed. Interestingly, irradiation of the gelator with UV-light induced a *trans*-to-*cis* or *cis*-to-*trans* isomerization depending on the wavelength (Scheme 1). Furthermore, only the *trans*-isomer acts as an efficient gelator, making this system an effective photoswitchable supramolecular gel. By studying the possible conformations of both the *trans-* and *cis*-isomers via DFT calculations, they concluded that the absence of gelation characteristics of the *cis*-isomer could be rationalized by the formation of an intramolecular hydrogen bond. In previous examples, explicit intramolecular interactions disrupted gel formation. However, we recently showed, through static DFT calculations, that, depending on the molecular structure, intramolecular hydrogen bonding does not necessarily lead to the loss of gelation characteristics [13].

Apart from interactions in and between the gelator molecules, static QM calculations can be used to gain information on the strength and nature of explicit interactions between the gelator and an additive or the solvent [40,43]. In a collaborative effort of the Rai and Kundu groups, a variety of structures of a solvent molecule and a di-Fmoc-*L*-lysine gelator were optimized at the DFT level of theory [44]. A variety of combinations of the solvent molecule and the gelator were built based on insights from the electrostatic potential maps of the gelator and were able to elucidate different contributions to the gelator–solvent interactions, such as explicit hydrogen bonding. In addition, these calculations allowed to determine a pseudo cohesive energy density (PCED) of the gelator–solvent mixture as a measure of the binding energy per unit volume. By normalizing this value with the CED of the pure solvent, a parameter Λ is obtained which quantifies the strength of the gelator–solvent interactions compared to the solvent–solvent interactions. Remarkably, their study shows that the value of Λ has a qualitative correlation with the solubility behavior of the gelator in different solvents.

### 2.2. All-Atom Molecular Mechanics and Dynamic Simulations

Molecular mechanics (MM) calculations are based on classical mechanics laws, with the relationship between the potential of a system and its topology described by a force field [45]. Because of the simplicity of most force field expressions, the computational workload of an MM calculation is significantly reduced, compared to a QM calculation on the exact same system. As can be expected, the performance of an MM calculation is highly dependent on the choice of force field, its parametrization and the accuracy thereof when describing the molecular system of interest. The fact that a multitude of different types of force fields exists, and more are constantly being developed, does not always make this easy for the end-user [46,47,48,49,50,51]. In this regard, special care must be taken when performing MM based calculations on supramolecular gels to ensure meaningful results. A proper choice of the force field and accurate parametrization can be obtained by comparison with experimental properties or computations performed at the QM level of theory.

Early research on supramolecular gels, in which MM computations were employed, featured static calculations as described in the previous section, involving single gelator molecules or small clusters to investigate possible stacking modes [52,53]. Nevertheless, as computational power increased and software was developed to take advantage of GPU-acceleration, molecular dynamic (MD) simulations of larger supramolecular aggregates in explicit solvation became accessible [54,55,56]. For supramolecular gels, this specifically means that, at the moment, nano- to microsecond simulations of an aggregate containing hundreds of small gelator molecules are within reach with a modest GPU-based computing cluster [57]. While this time and length scale are still far too small to simulate and unveil the full self-assembly process taking place during gelation, two approaches currently exist to study this process by means of MD simulations: a top-down or a bottom-up approach (Figure 3) [15].

In a top-down approach, the MD simulation starts from an organized supramolecular aggregate, of which the stability is probed by analyzing the topological changes occurring during the simulation. It is recommended that, instead of one, several properties are used to quantify the topological changes, such as the root-mean-square displacement of all atoms (RMSD), the solvent accessible surface area (SASA), the radius of gyration (R_g_), or other geometric parameters that might be unique for the aggregate [58]. If the proposed structure is unstable, atoms will diverge from one another and a reorganization of the structure will become apparent. On the other hand, if the proposed structure is stable, the relative position of the atoms in the structure will have little displacement along the trajectory and an equilibrated structure will be reached. Only supramolecular structures with a certain degree of stability are then considered to analyze plausible stacking modes that might be important during gelation [59,60,61,62,63,64,65,66,67]. The topology of the initial proposed structure is crucial to the value of this method. Moreover, while it can be challenging to propose an adequate initial structure based on chemical intuition, experimental data on the morphology of the gel network can help narrow down the possibilities and provide the necessary empirical validation.

In recent research published by the Steed group, a top-down MD approach was used to better understand how unique twisting ribbons were formed during the self-assembly of a pentakis(urea) based supramolecular gelator [68]. From TEM imaging, it was established that the gelator self-assembles into symmetrical twisted ribbons with a width of approximately 16–18 nm. Based on this experimental observation, a stacking model was constructed by having six 1-dimensional arrays of gelators with a helical pitch of 40 nm. Crucially, this stacking model complied with the experimental morphological observations. Next, fibrils with 2, 3 or 4 of these layers were simulated for 2.5 ns. From these simulations it became clear that only a structure containing 4 layers would be sufficiently stable during a 2.5 ns simulation (Figure 4). In this study, a conventional top-down MD approach was employed. However, in other cases, the proposed structure undergoes a significant reorganization. The same group used MD simulations in a previous study to investigate the self-assembly of a bis-urea based gelator [69]. Starting from an organized pre-stacked lamellar structure of the gelator, scrolling towards a cylindrical structure was observed. In another example, the Pavan group in collaboration with the Meier group used a top-down MD approach to elucidate the self-assembly of a 1,3,5-benzenetricarboxamide (BTA) derivative in water towards a supramolecular fiber [70]. The proposed aggregate for the simulation was built by sequential addition of pre-stacked structures that were optimized using DFT calculations and consisted of 48 molecules. The resulting fiber was placed in explicit water (TIP3P model) and underwent a dynamic simulation of 400 ns. Notably, two folding stages were observed before an equilibrium structure was reached. In a first stage, the side chains of the BTA cores collapse around the core of the fiber to reduce the hydrophobic interactions between the fiber and the solvent, while in a second stage the fiber itself begins to fold, further minimizing the hydrophobic interactions. Only after 300 ns, an equilibrium structure is reached, underlining the importance of an adequate total simulation time when using a top-down MD approach.

In a bottom-up MD approach, the simulation starts from a random topology, preferably with the gelator molecules dispersed across the entire solvent medium. While computational limitations currently render it unfeasible to reach an equilibrated state using this approach, valuable insights can still be gained regarding the early self-assembly stage and specific non-covalent interactions driving supramolecular gelation [13,71]. While fewer of such studies exist, the Marlow and Zelzer group used a bottom-up MD approach to investigate the initial self-assembly of a nucleoside based gel at two different concentrations in an ethanol:water (20:80 *v*/*v*%) mixture (Figure 5) [72]. At lower concentrations, smaller aggregates of the gelator were formed based on a parallel packing to maximize π–π interactions. They also observed that the solvation around these aggregates was not homogeneous with the aliphatic tails mainly solvated by ethanol molecules, while the cytosine bases were mostly solvated by water. Prolongation of the simulation resulted in the reordering of these aggregates into a single flexible aggregate. When the concentration of the gelator was increased in the simulation box, similar results were obtained, although, in this case, an aggregate was formed that stretched across the periodic boundary of the box. In another study using a bottom-up MD approach, the Seddon and Adams groups investigated the packing of a small peptide based gelator (NapFF) in water [73]. More specifically, biased MD simulations starting from a dispersed state of NapFF molecules in water resulted in the formation of a hollow tubular aggregate of gelators. In this study, cylindrical restraints were introduced during the simulation to obtain an aggregate which would be consistent with the experimental scattering data. Upon relieving the restraints, the tubular structure did remain stable, due to the Na^+^ ions present in the simulation box and hydrogen bonding within the structure.

### 2.3. United-Atom and Coarse-Grained Simulations

The previously highlighted study showcases that geometrical constraints can be used to speed-up the self-assembly process during a bottom-up MD simulation. It is important, however, that these restraints are made based on a sound rationale. Another workaround to deal with limited computational power during a simulation is simplifying the force field that drives the dynamics. Indeed, the previously mentioned dynamic studies all used an all-atom force field expression to describe the gelators, in which each atom is explicitly parametrized. Simplified force field expressions can be obtained if multiple atoms are combined and treated as a single interacting site. One of the most popular methods that uses this approach is the united-atom potential, in which the hydrogen and carbon atoms of methyl or methylene groups are combined [74,75,76,77,78,79]. In addition, when describing supramolecular gels, a united-atom force field can be useful to increase the size- and time-scale of the simulation [80]. In a recent example, the Rai group used the transferable potentials for phase equilibria-united atom (TraPPE-UA) force field to investigate the aggregation of 12-hydroxyoctadecanamide in octane [77,81]. Thanks to the united-atom force field, they were able to run bottom-up MD simulations on 500–1000 gelators in explicit octane (12.5 wt%) for 500 ns. With this time and length scale of simulation, they were able to observe the initial stage of fiber branching and reveal the importance of the hydroxyl group in the branching process.

More simplified potentials also exist, in which multiple atoms, depending on the functional group, are combined to a single interacting site. These potentials are referred to as coarse-grained (CG) force fields [82,83]. It is important to realize that the development of these CG potentials requires extensive efforts to ensure an adequate accuracy when describing the system, despite the significant loss of detail. In the case of supramolecular gels and biomolecular systems, the MARTINI CG force field is widely used due to its extensive development for the description of amino acids [84,85]. In 2012, the Schatz group used the MARTINI force field to model the self-assembly of the Ile-Lys-Val-Ala-Val (IKVAV) sequenced peptide amphiphile [86]. The CG MD simulations could reach up to several microseconds, which allowed them to observe spontaneous cylindrical fiber formation starting from a fully dispersed state of the peptides. Similarly, the Wei and Gazit groups made use of the MARTINI force field to investigate the self-assembly of a small peptide-based hydrogelator [87]. Similarly to the results obtained by Schatz and co-workers, the microseconds-long CG MD simulations enabled them to observe the self-assembly towards wormlike aggregates starting from a dispersed state. One obvious limitation to these approaches, however, is that due to the loss of atomistic details, it is not possible to directly correlate the self-assembly process to specific non-covalent interactions. To overcome this limitation, Nguyen and co-workers developed ePRIME, a CG model with intermediate resolution that combines enough detail to obtain insights into specific interactions, while pertaining computational tractability for the simulation of large systems [88]. In 2013, they used the ePRIME model to investigate the spontaneous self-assembly of peptide amphiphiles and to successfully construct phase diagrams that delineate morphological transitions triggered by external stimuli such as temperature [89]. Alternatively, one could opt to perform hybrid-resolution simulations, in which part of the system is described by a CG model, while other (more essential) parts are described by a united- or all-atom force field [90]. Finally, we note that some methods have been developed to retrieve the atomistic picture from the CG representation; however, they have yet to be tested in the field [91].

### 2.4. Other Methods

In addition to the archetypical static and dynamic computations mentioned above, other less conventional computational methods can be employed to gain valuable insights into the supramolecular gelation process. One such example is the Non-Covalent Interactions index (NCI). Developed in 2010 by the Yang group, the NCI allows for a 3-dimensional visualization of all non-covalent interactions in a molecular system, together with an indication of their strength [92,93,94,95,96]. While a full theoretical description of the NCI technique falls outside the scope of this review, it is important to recognize that the method can pinpoint noncovalent interactions in supramolecular gels, revealing their importance during gelation [13,97,98]. Briefly, the NCI method is based on the reduced density gradient and allows for the characterization and visualization of noncovalent interactions in real space. With current density functionals suffering from severe limitations in terms of generality for describing different types of noncovalent interactions, the NCI is extremely robust with respect to the computational method [99]. Indeed, some of the authors of this work have proven that the NCI method outperforms conventional methods to establish the hydrogen-bond network in proteins [100,101].

## 3. Predicting Supramolecular Gelation

From previous paragraphs, it became clear that computational tools can be used to study the self-assembly process that occurs during supramolecular gelation. Besides rationalization, researchers have dedicated their attention towards accelerating the discovery of novel supramolecular gels through computations. As a consequence, different methods have been proposed to predict the gelation propensity of a molecule in a solvent. While these methods differ in approach and applicability, they are all driven by computations and realize a reduction of chemical space, hence simplifying the selection of potential molecules that can gel a solvent.

### 3.1. Predicition through the Crystal Structure

In many ways, supramolecular gelation of a molecule in a solvent is related to its crystallization. Both supramolecular gelation and crystallization are non-equilibrium self-assembly processes that occur under supersaturated conditions and are characterized by a nucleation and growth phase [102,103]. For this reason, many experimental studies on supramolecular gels report powder X-ray diffraction data of the dried xerogel with the aim of extracting useful information on the structure of the gel network [5,104,105]. From the 1960s onwards, crystal structure prediction (CSP) has become a field on its own and has experienced an enormous development of applicable techniques able to predict the crystal morphology based on the molecular structure [106,107,108,109,110]. One strategy that has been explored to accelerate the design of new supramolecular gels is based on predicting the crystal structure of the gelator through a CSP method and, subsequently, analyzing if properties found in the predicted crystal morphology correlate to the gelation propensity of the molecule [111,112,113,114,115].

In 2016, the McNeil group successfully developed the first Pb-containing supramolecular gelators based on a method exploiting CSP (Figure 6) [116]. Starting from all Pb-containing crystals available in the Cambridge Structural Database (CSD), a set of rational filters was applied to narrow down the scope to 352 structures. In the next step, molecular mechanics driven geometry optimizations further reduced the possible candidates to 184, which formed a set of workable size. Through generating a crystal graph for each compound, describing the interaction energy between the center of mass of one molecule with all other molecules in a preset sphere, the crystal morphology of all 184 compounds was predicted using the Growth Morphology software within Materials Studio [117]. After this, the aspect ratio was computed for all predicted crystal morphologies, by taking the ratio of the longest distance within the crystal to the shortest distance. Finally, two compounds were selected for derivatization and gel screening, by looking at the predicted crystal structures with the highest aspect ratio (top 5%) and with ease of chemical synthesis in mind. As a result of this approach, a first class of 6 Pb-containing supramolecular gelators was designed, out of which one was explored for its ability to sense Pb^2+^ in paint.

While the work of the McNeil group showcases the potential of using CSP to accelerate the development of supramolecular gelators, we do want to emphasize that one must be critical when scrutinizing data related to the crystal structure to draw conclusions on the corresponding supramolecular gelation process. Albeit the crystalline- and supramolecular gelation process are related to some extent, it has been pointed out, on several occasions, that the crystal structure of a molecule can significantly differ from the stacking mode and noncovalent interactions present in its gel phase [118,119,120].

### 3.2. Solvent Parameters

In most supramolecular gels, the solvent makes up 99% of the material or even more. Evidently, one can expect that the properties of the supramolecular gel are highly dependent on the solvent. As a consequence, many groups have analyzed possible correlations between gel propensity or properties of the gel phase and intrinsic parameters of the solvent, such as the solvent dielectric constant, polarity parameter E_T_(30), the Kamlet–Taft solvent parameters or the Hildebrand solubility parameter [121,122,123,124,125,126,127]. Over the years, the Hanssen solubility parameters (HSPs) have proven to be highly useful when investigating complex solvation effects or predicting gelable solvents [128,129,130,131]. Briefly, the HSPs are a result from the decomposition of the Hildebrand solubility parameter into three contributing molecular interactions: dispersive interactions (*δ*_d_), polar interactions (*δ*_p_) and hydrogen-bonding interactions (*δ*_h_) [132]. Originally intended for selecting solvents for polymeric systems, the Bouteiller group was the first to successfully manipulate the HSPs to predict gelable solvents for a specific supramolecular gelator [133,134,135]. In their approach, an initial solubility dataset is constructed by testing the gelation behavior of a gelator in various solvents. Results are classified into three categories: the molecule is soluble in the solvent (S), gels the solvent (G), or forms a precipitate (P). Based on this dataset, a solubility sphere and one or more gelation spheres are defined in the Hansen space. The radius and center of these spheres are determined by an optimization which results in having most S points inside the solubility sphere but outside the gelation sphere (s), most G points inside the gelation sphere(s) but outside the solubility sphere and most P points outside both the solubility and gelation spheres. In this manner, the solubility behavior of the gelator in an untested solvent can be easily predicted based on the location of the solvent in Hansen space.

Further investigations revealed that the method described above could be improved and simplified by allowing a single solubility and gelation sphere to overlap (Figure 7) [129,136,137,138]. The Rogers group had a notable impact in further development of this methodology [139,140,141]. In a more recent work from the group, the values of the radius and center of the constructed gelation sphere were used to elucidate the ability and driving forces behind the gelation of different peptide-based supramolecular gelators [142]. The gelation sphere obtained for *L*-diphenylalanine (*L*-FF) was centered around the values of 2*δ*_d_ = 32.80 MPa^1/2^, *δ*_p_ = 8.70 MPa^1/2^ and *δ*_h_ = 11.50 MPa^1/2^ and had a radius of 5.61 MPa^1/2^. On the other hand, the gelation sphere for *L*-dityrosine (*L*-YY) was centered around 2*δ*_d_ = 31.39 MPa^1/2^, *δ*_p_ = 15.75 MPa^1/2^ and *δ*_h_ = 14.65 MPa^1/2^ and had a radius of 18.50 MPa^1/2^. The increased radius of the gelation sphere of *L*-YY in comparison to *L*-FF, highlights the greater gelation capacity of *L*-YY. Moreover, the shift of the gelation sphere towards more polar and greater hydrogen bonding solvents when going from *L*-FF to *L*-YY underlines the change in the driving force behind gelation towards polar interactions and hydrogen bonding upon including additional hydroxyl groups to the structure.

### 3.3. Molecular Dynamics and Machine Learning

Whilst the value of the HSP method described above is obvious from the many studies on supramolecular gels that rely on the technique, it does require extensive laborious efforts, as a large collection of experimental gelation data needs to be gathered beforehand. To further accelerate the discovery of new potential gelators, it would be beneficial to be able to predict the gelation propensity of a molecule before stepping into the lab. In this section, a number of strategies are highlighted that can evaluate the gelation propensity of a molecule, solely based on the molecular structure of the gelator.

A significant contribution towards the prediction of supramolecular gelation was made by the collaborative efforts from the Tuttle and Ulijn groups [143]. Already in 2011, they proposed an aggregation propensity score (AP) able to quantify the aggregation tendency of a given peptide [144]. More specifically, starting from a fully dispersed state, coarse-grained simulations are performed to track the self-assembly behavior of a small peptide. Next, the AP score is computed by taking the ratio of the solvent accessible surface area (SASA) of the peptides at the initial topology to the SASA of the peptides at the end of the simulation. An AP value greater than 2 is arbitrarily proposed as an indication of a high degree of aggregation. Four years later, they reported a corrected AP score (AP_H_), which incorporates the hydrophilicity of the peptide by multiplying the original AP score with a computed log P value [145]. Notably, they screened over 8000 tripeptides and successfully showed that this hydrophilicity-corrected AP_H_ score can be used as a reliable descriptor to create a set of design rules for the development of new peptide-based hydrogelators. We believe the success of their approach can largely be attributed to the physical relevance of the proposed descriptor towards describing the self-assembly process.

Another approach to predict supramolecular gelation relies on machine learning schemes. Based on collected experimental data and a large descriptor set, the Adams and Berry group were the first to successfully train a number of machine learning models able to accurately classify dipeptide systems into efficient hydrogelators and non-hydrogelators [146]. Remarkably, the prediction is based on a variety of physicochemical properties and molecular fingerprints that are generated based on the simplified molecular-input line-entry specification (SMILES) representation of the gelator. This allowed for a high-throughput prediction, as long as the SMILES code falls within the applicability domain of the model. While chemical intuition does not initially link the descriptors used to build the model to the gelation process, the work is an excellent example of how different machine learning techniques can exploit large amounts of data and retrieve complex relationships between them. In addition, the Fok and Li groups adapted a similar machine learning approach to predict the gelation propensity of dipeptide hydrogelators [147].

Recently, our group developed a set of descriptors, which can all be obtained from all-atom molecular dynamics simulations [148]. The relative solvent accessible surface area (rSASA) quantifies aggregation of the gelators, the relative end-to-end distance (rH) describes the flexibility and conformational preference of the gelators, the hydrogen bonding percentage (HB%) measures the noncovalent linkage between gelator molecules through hydrogen bonding and the shape factor (F) describes the aggregate’s shape. These descriptors are of fundamental value to the gelation process, but more importantly they can be used to optimize machine learning models that are able to classify combinations of a urea-based gelator and a number of solvents into three categories. Either the gelator is soluble in the solvent, forms a precipitate or gels the solvent (Figure 8). Notably, this is the same classification that is obtained from the HSP method described earlier. Due to their relevance in describing the self-assembly that occurs during the simulation, we anticipate these descriptors to be valuable for different types of supramolecular gelators. One drawback to the method, however, is that extensive MD simulations need to be performed to obtain the descriptor, limiting the throughput of prediction.

## 4. Conclusions and Future Perspectives

In this review, we discussed the potential of computational methods within the field of supramolecular gels. A thorough examination of the studies performed on supramolecular gels in the past two decades revealed that computational methods can improve our fundamental understanding of the self-assembly behavior of supramolecular gels. Different computational techniques were introduced that can provide insights on different time and length scales of the supramolecular gelation process. A combination of multiple computational methods integrated into a multiscale approach provides the most complete picture, complementary to well-established wet-lab experiments. In addition to rationalization, computational tools have shown the ability to accelerate the development of supramolecular gels, by defining boundaries around a specific area of chemical space and simplifying the selection of possible gelator candidates through various approaches. This review attempts to create awareness on the added value of computational chemistry when investigating supramolecular gels, as well as provide an overview of the current possibilities and state-of-the-art.

As the field moves forward, we believe the use of computational methods will become even more prominent when studying supramolecular gels. In the upcoming years, the ever-increasing computing power will enable researchers to perform calculations with an accuracy that was previously considered unattainable [149]. For supramolecular gels specifically, this could mean investigating larger molecular events, such as fiber entanglement, with atomistic precision. Another challenge the field faces is creating a more efficient discovery process for supramolecular gels with real-life use. A variety of potential applications are explored for these soft materials, with each application requiring specific material properties. Obtaining *a priori* knowledge on the tunability of these material properties, would drastically change the development strategy of supramolecular gels and could enable them to have a tangible impact on society. One approach, currently being heavily explored for the directed design of other next generation materials, is establishing quantitative structure–property relationships by means of machine learning [150,151,152,153,154,155]. In previous paragraphs, we already touched on the use of machine learning models to facilitate the discovery of gelators [146,147,148]. In this regard, collecting material properties of different supramolecular gels that are obtained through a consistent and unambiguous manner in a central and open database would greatly expedite the search for quantitative structure–property relationships through machine learning.

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
