# Peer review of "Computational Tools to Rationalize and Predict the Self-Assembly Behavior of Supramolecular Gels"

_gels, 2021, doi:10.3390/gels7030087_

Round 1

Reviewer 1 Report

In this review, the authors summarize the progress in application of computational approaches to rationalize self-assembly behavior of supramolecular gels and assist in their design. The review is well organized and written. It covers a wide range of contents concerning why and how various computational approaches could be applied to tackle different scientific questions pertaining to self-assembly mechanism of molecular gels and their design principle. These contents are in general timely and comprehensive, representing the state-of-the-art progress in the field. This review would provide valuable guidance to readers who are interested in employing computational approaches to study self-assembly of molecular gels and assist in their rational design. Therefore, I recommend this manuscript for publication in Gels if my following minor comment is addressed.

In section 2.3, the authors discussed the applications and limitations of united-atom and coarse-grained models in modeling self-assembly of gels, with the former models being computationally demanding and the latter models lacking in important details. Nonetheless, there are several types of computational models that have been developed to tackle, at least partially, these limitations, such as the models with an intermediate resolution (Adv Healthcare Mater 2013, 2, 1388) and the models that combine united-atom representation and coarse-grained representation together (ACS Nano 2019, 13, 4455). The authors may consider covering these works in their review.

Reviewer 2 Report

This is a comprehensive review on the available computational methods to study supramolecular gels. The manuscript is clearly written with an excellent coverage of the available literature on the subject. The different approaches for rationalizaing gelating properties are introduced first with a strong emphasis in quantum mechanical methods (mainly DFT). Then methods to predict gelating properties mainly based in SAR and QSAR type approaches are presented. This review will be useful to a broad audience as it is written in a way that may be easily followed by the non-specialist. I recommend publication.

Reviewer 3 Report

This manuscript reviews recent computational studies to rationalize and predict the self-assembling behaviors of the supramolecular gels. A class of the supramolecular gels is one of the most attracted soft matters and, therefore, experimental researchers have made efforts to develop new supramolecular-gel system on the basis of a try-and-error strategy and structure-property relationship. Recent progress on the computational study is remarkable, and it must be helpful in the field of the supramolecular gels as well. In this review, recent and balanced reports are highlighted and, therefore, it is informative for the broad readers of the Journal. Thus, the reviewer recommends the manuscript is acceptable. 
